# Co-Exposure of Polycyclic Aromatic Hydrocarbons and Phthalates with Blood Cell-Based Inflammation in Early Pregnant Women

**DOI:** 10.3390/toxics11100810

**Published:** 2023-09-25

**Authors:** Yunxiao Yang, Ting Wang, Lei Luo, Qian He, Fangfei Guo, Zhongbao Chen, Yijun Liu, Xingyan Liu, Yan Xie, Xuejun Shang, Xubo Shen, Yuanzhong Zhou, Kunming Tian

**Affiliations:** 1School of Nursing, Zunyi Medical University, Zunyi 563000, China; 13368679346@163.com; 2School of Public Health, Zunyi Medical University, Zunyi 563000, China; dr_luolei7007@163.com; 3School of Health Policy and Management, Nanjing Medical University, Nanjing 210000, China; 17881600860@163.com; 4Department of Obstetrics and Gynecology, Affiliated Hospital of Zunyi Medical University, Zunyi 563000, China; hq1040630015@163.com; 5School of Public Health, Guizhou Medical University, Guiyang 550000, China; 18311841914@163.com; 6Renhuai Center for Disease Control and Prevention, Zunyi 563000, China; chenzhongbao0128@163.com; 7Key Laboratory of Maternal & Child Health, Exposure Science of Guizhou Higher Education Institutes, Zunyi 563000, China; liuyijun_0422@126.com (Y.L.); sprite95@126.com (X.L.); xie814yan@163.com (Y.X.); zy96_shenxubo@163.com (X.S.); 8Department of Andrology, School of Medicine, Jinling Hospital, Nanjing University, Nanjing 210002, China; shangxj98@sina.com

**Keywords:** pregnancy, BKMR, Q-g, Inflammation, PAHs, PAEs

## Abstract

Cumulative evidence has demonstrated that exposure to polycyclic aromatic hydrocarbons (PAHs) or phthalates (PAEs) contributes to a variety of adverse health effects. However, the association of PAHs and PAEs co-exposure with blood cell-based inflammatory indicators during early pregnancy is still unclear. We aimed to investigate the single and mixed associations of exposure to PAHs and PAEs with blood cell-based inflammatory indicators among early pregnant women. A total of 318 early pregnant women were included in this study. General linear regressions were used to estimate the relationships of individual OH-PAHs and mPAEs with blood cell-based inflammatory indicators. The key pollutants were selected by an adapted least absolute shrinkage and selection operator (LASSO) penalized regression model and wasemployed to build the Bayesian kernel machine regression (BKMR) and quantile g-computation (Q-g) models, which can assess the joint association of OH-PAHs and mPAEs with blood cell-based inflammatory indicators. General linear regression indicated that each 1% increase in MOP was associated with a 4.92% (95% CI: 2.12%, 7.68%), 3.25% (95% CI: 0.50%, 6.18%), 5.87% (95% CI: 2.22%, 9.64%), and 6.50% (95% CI: 3.46%, 9.64%) increase in WBC, lymphocytes, neutrophils, and monocytes, respectively. BKMR and Q-g analysis showed that the mixture of OH-PAHs and mPAEs was linked with increased levels of white blood cells (WBC), neutrophils, monocytes, and lymphocytes, and MOP was identified as the dominant contributor. OH-PAHs and mPAEs co-exposure in early pregnancy was associated with elevated blood cell-based inflammatory indicators reactions. More attention should be paid to the inflammation induced by environmental pollution for perinatal women, especially early pregnant women.

## 1. Introduction

Polycyclic aromatic hydrocarbons (PAHs) and phthalates (PAEs) are organic pollutants ubiquitously present in the environment, mainly created by transportation, home heating, industrial emissions, and biomass burning (including slash-and-burn agriculture) [1,2]. It should be noted that the annual supply of PAEs rose from 2.7 to over 6 million tons globally between 2007 and 2017 [1,3], the majority of which were consumed by developing countries [1]. Exposure to PAHs or PAEs has been suggested to drive various detrimental health effects, including abnormal lung function, atherosclerosis, myocardial infarction, and various cancers [4,5,6,7]. Therefore, it is essential to study the adverse health effects resulting from PAHs and/or PAEs exposure.

A normal inflammatory reaction is critical to maintaining the homeostasis of the human body, and the pathological inflammatory reaction is closely associated with the initiation and progression of many diseases [8,9]. During the perinatal period, abnormal maternal inflammatory response often leads to spontaneous abortion, preterm delivery, and fetal malformations, even neurodevelopmental disorders in childhood [10,11]. Thus, focusing on inflammatory levels to prevent negative effects is important for mothers and newborns. Blood-based inflammatory indicators have been widely used in the clinical setting, including the white blood cell (WBC), lymphocyte, neutrophil, monocyte, platelet, neutrophil-to-lymphocyte ratio (NLR), platelet-t1 mL o-lymphocyte ratio (PLR), and systemic immune-inflammation index (SII). It is currently believed that using these indicators to assess biochemical impacts, and the human body’s inflammatory and immunological condition is preferable [12,13], due to their great repeatability and low-price characteristics [14]. 

Increasing evidence indicated that increased circulating concentrations of these inflammatory indicators derived from air pollution are partially attributed to the PAHs and PAEs in PM2.5 [8,15]. Exposure to PAHs or PAEs has been linked with altered inflammatory markers [12,16], whereas these results remain inconsistent. Co-exposure to PAHs and PAEs was linked with raised inflammatory indicators of children aged 4 to 12 in Guangzhou (China), and ∑ OHPHE and 1-OHPYR are the main chemicals [12]. Similarly, urinary PAHs concentrations were positively associated with serum CRP and total WBC levels among adults in the United States [17]. However, several inflammatory indices, such as WBC and platelet count (PLT), were negatively correlated with urinary PAEs metabolites, but positively associated with certain platelet parameters including PLR, mean platelet volume (MPV), and platelet distribution width (PDW) among the Chinese population [18]. Therefore, it is important to capture the health effects related to PAEs or PAHs among different populations.

Previous studies commonly determined that these health effects resulted from single exposure of PAHs or PAEs to perinatal women. However, to the best of our knowledge, there has been no research determining the relationship of combined exposure to PAHs and PAEs with blood cell-based inflammatory indicators among early pregnant women. Furthermore, early pregnancy is a vulnerable period to exposure to environmental factors exposure and a vital period for fetal growth. Therefore, we aimed to comprehensively examine the individual and joint association of PAHs and PAEs with blood cell-based inflammatory indicators during early pregnancy.

## 2. Methods

### 2.1. Study Population

We conducted a multi-center hospital-based Zunyi birth cohort from May 2020 to April 2022. This cohort included women in early pregnancy who were confirmed pregnant and free from serious chronic or infectious diseases (e.g., cancer, chronic cardiovascular disease, chronic renal failure, and HIV infection). A total of 393 early pregnant women with both urine and blood samples were initially included in the study, and we then excluded pregnant women who were definitively diagnosed by their physicians as having an inflammation-related disease (*n* = 75). Finally, 318 pregnant women were eligible to participate in this study. We compared the basic characteristics of the included (*n* = 318) and initial (*n* = 393) groups and found no significant differences (Appendix A). Face-to-face interviews were conducted to collect basic information, and all women provided informed consent. 

### 2.2. Questionnaire Survey

Covariates were selected on the basis of possible biological mechanisms, available evidence, and statistical stepwise regression [12,19]. The questionnaire contains information about height, weight, age, ethnicity, marital status, history of active or passive smoking, alcohol use, daily activities, etc. Height and weight were obtained via medical records and were used to calculate pre-pregnancy body mass index (BMI) (kg/m^2^). Women of different ethnicities were divided into two groups (ethnic minorities and ethnic Han); education was divided between a college degree or above and below a college degree; participants’ marital status was divided into three categories: single, married, and divorced. Exercise conditions were divided into three groups (<1 time/week, 1–2 times/week, ≥3 times/week); passive smoking is defined as passive exposure to cigarettes from the workplace, home, or public places at least once a month during pregnancy; participants were divided into two groups based on whether they smoked or drank alcohol.

### 2.3. Determinations of Urinary mPAEs and OH-PAHs

Urine samples collected from women were used to examine the PAEs and PAHs. The PAH metabolites are mono-methyl phthalate (MMP), mono-ethyl phthalate (MEP), mono-isobutyl phthalate (MIBP), monobutyl phthalate (MBP), mono-octyl phthalate (MOP), monobenzyl phthalate (MBZP), mono (2-ethylhexyl) phthalate (MEHP), mono (2-ethyl-5-oxohexyl) phthalate (MEOHP), mono (5-carboxy-2-ethylpentyl) phthalate (MECPP), and mono (2-ethyl-5-hydroxyhexyl) phthalate (MEHHP), respectively. The PAH metabolites were respectively 1-hydroxynaphthalene (1-OHNAP), 2-hydroxynaphthalene (2-OHNAP), 2-hydroxy fluorene (2-OH-FLU), 9-hydroxy fluorene (9-OHFLU), 1-hydroxyphenanthrene (1-OH-PHE), 2-hydroxyphenanthrene (2-OH-PHE), 3-hydroxyphenanthrene (3-OH-PHE), 4-hydroxyphenanthrene (4-OH-PHE), 9-hydroxyphenanthrene (9-OH-PHE), and 1-hydroxylysine (1-OH-PYR), respectively. 

Briefly, 1 mL of sodium acetate buffer solution, 10 μL of internal standard solution, and 10 μL β-glucuronidase/sulfatase were added to 1.5 mL of urine sample. It was then placed in water at 37 °C for 12–16 h. The digested material was mixed with MgSO_4.7_H_2_O, followed by hexane-ether solvents (4:1) for extraction. Prior to being centrifuged for 10 min (3500 rpm), the solution was blended for 30 s using the multi-tube vortex mixer. The chromatographic inlet port was set at split less mode, the carrier gas was 99.999% helium flowing at a rate of 1.2 mL/min, and the injection volume was 1 L [20]. The completed product was centrifuged to separate and concentrate the organic phase using a dry nitrogen stream. An equal amount (100 μL) of silylation reagent was added, and the vials were tightly covered and then submitted to a water bath (temperature set at 90 °C for 45 min) to allow sufficient derivatization, cooled, and finally analyzed using a gas chromatograph-triple quadrupole mass spectrometer (GC-MS/MS, Agilent 7010b; Agilent Technologies, Inc; Santa Clara, CA, USA) [20,21,22].

### 2.4. Quality Control

Polypropylene tubes were used to collect random urine samples, which was conducted on the same day as the blood samples collection. The standard addition method was used; namely, 1.5 mL of urine was added to three different concentrations of standard solutions 3.8%—low, medium, and high. Each concentration was in eight parallel tubes, and the recovery and detection precision were computed. The signal-to-noise ratio (S/N) of 3 was chosen as the limit of detection (LOD). There was at least one experimental blank in each batch of 20 urine samples, and the relative deviation of two parallel samples should be less than or equal to 20% to account for the background value that the experimental procedure and reagents must always account for. In order to prevent instrument bias or other interferences, we also monitored the intensity of the internal standard in each test. The internal standard’s response value should be within 30% of the calibration curve’s response value. The retention time, ion pair, collision energy, recovery rate, precision, determination coefficient, and limit of detection were presented in Appendix A.

### 2.5. Blood Cell Examination

The blood and urine samples were collected on the same day women were enrolled. A venous blood sample was collected using a siliconized vacutainer tube with 3.8% sodium citrate (volume ratio 1:9) in the morning after an overnight fast. Serum samples were centrifuged within 2 h and the WBC, lymphocyte, neutrophil, monocyte, and platelet counts were determined using a BC-30 automated hematology analyzer (Mindray; Shenzhen, China). All the hemocyte concentration was expressed as units of 10^9^/L. NLR stands for neutrophil/lymphocyte ratio, whereas PLR refers to platelet/lymphocyte ratio. Neutrophil count times platelet count divided by lymphocyte count was used to determine SII.

### 2.6. Statistical Analyses

The distribution of data was obtained by the Kolmogorov-Smirnov test. Normally distributed data were presented as mean ± standard deviation (SD), and categorical covariates were expressed as percentages.

The concentration below the limit of detection (LOD) was calculated as LOD/√2 [23]. To account for urine dilution, the urinary OH-PAHs and mPAEs concentrations were corrected using the following formula: urinary creativity correction phthalate metabolite concentration (μg/g) = phthalate metabolite concentration (μg/L)/[urinary creatinine concentration (μg/L) × 113 (g/mol) × 10^−6^]. The concentrations of urinary OH-PAHs and mPAEs, as well as the values of the inflammatory markers, were naturally ln-transformed due to skewed distributions. Spearman rank correlation was employed to assess the relationship among chemicals. To control the multicollinearity of 19 pollutants, an adapted least absolute shrinkage and selection operator (LASSO) penalized regression model was used to determine the major driver of WBC, and 10-fold cross-validation was carried out to establish the smallest value of λ. WBC was selected as a dependent variable for roommate screening based on the following factors: (1) WBC activation and translocation are critical in the initiation of inflammatory reaction and fighting inflammation [24,25]. (2) WBC is one of three dominant blood cell types, and they play various crucial physiological roles; for instance, phagocytosis (by lymphocytes) can eliminate pathogens through phagocytosis (by neutrophils) and by converting into macrophages (by monocytes) [26,27]. (3) Notably, elevated leukocytes can represent a normal response to infection [26,28].

The relationship between a single exposure to mPAEs or OH-PAHs and blood-cell-based inflammatory indicators was examined using linear regression models. Furthermore, considering the potential nonlinear and synergistic effects among the above-selected contaminants, a Bayesian kernel machine regression (BKMR) model with 50,000 iterations was built to explore the overall effects of various pollutants. The combined association of chemicals with inflammatory indictors was also validated by quantile g (Q-g) calculations. We excluded women with aberrant WBC < 4.0 × 10^9^/L or > 12.0 × 10^9^/L as a sensitivity analysis to examine the robustness of the results [12].

SPSS version 24.0 (IBM Corp; Armonk, NY, USA) and R (version 4.1.0; Integrated R Archive Network, New Zealand) were performed to analyze our data. Two-tailed *p* < 0.05 was considered a statistical significance.

## 3. Results

### 3.1. General Characteristics of the Participants

The baseline characteristics of 318 first-trimester pregnant women were shown in Table 1. Average age was 26.22 ± 4.7 years old, the pre-pregnancy BMI was 22.10 ± 4.22, 312 (98.11%) were ethnic Han, 232 (72.95%) were high school or below, 277 (87.11%) were married, 39 (12.26%) were single, and 2 (0.62%) were divorced; 20 (6.41%) women smoked, 313 (98.43%) had passive smoking. More than half of the pregnant women, 167 (52.52%), did little or no exercise during pregnancy, 41 (12.89%) exercised 1–2 times per week, and 110 (34.59%) exercised more than 3 times per week; 27 had a history of alcohol use.

### 3.2. Concentrations of Urine mPAEs, OH-PAHs, and Blood Cells

The inflammatory indicator distributions of 318 pregnant women were shown as below: WBC 8.09 (6.80, 9.58), lymphocyte 1.80 (1.50, 2.18), neutrophil 5.73 (4.60, 6.94), monocyte 0.40 (0.32, 0.48), platelet 241.5 (202, 274), NLR 3.11 (2.43, 3.94), PLR 130.50 (109.44, 155.56), and SII 727.00 (552.05, 942.74). As shown in Table 2, the detection rates of mPAHs and PAHs in urine were more than 70%, while the detection rates of 4-OHPH, MMP, MOP, and MBZP were lower than 70%, and MBZP (40.25%) was excluded from this study due to the detection rate < 60%.

### 3.3. Correlation between OH-PAHs and mPAEs Metabolites

The Spearman rank correlation was employed to examine the correlation between OH-PAHs and mPAEs metabolites (Figure 1). The positive correlations between the metabolites of various OH-PAHs and mPAEs ranged from 0.01 (MMP and 3-OHPH) to 0.79 (MIBP and MBP) (all *p* < 0.05). Significant negative correlations between 2-OHNAP and MEHP, MECPP and MEHP, MEOHP and 1-PHPYR, and 1-OHPYR and 2-OHPH were also found (all *p* < 0.05).

### 3.4. The Relationship between OH-PAHs and mPAEs and Blood Cell-Based Inflammatory Indicators

As shown in Figure 2, after adjusting for potential confounding factors, each 1% increase in MOP was associated with a 4.92% (95% CI: 2.12%, 7.68%), 3.25% (95% CI: 0.50%, 6.18%), 5.87% (95% CI: 2.22%, 9.64%), and 6.50% (95% CI: 3.46%, 9.64%) increase in WBC, lymphocytes, neutrophils, and monocytes, respectively; each 1% increase in MEOHP was associated with a 1.82% (95% CI: 0.10%, 3.56%) and 2.43% (95% CI: 0.20%, 4.71%) increase in WBC and neutrophils.

In addition, we also found that each1% increase in 1-OHNAP was associated with a 1.31% (95% CI: 0.40%, 2.33%), 1.82% (95% CI: 0.50%, 3.15%), 0.90 (95% CI: 0.10%, 1.82%), 1.61% (95% CI: 0.030%, 3.15%), and 2.53% (95% CI: 0.80%, 4.39%) increase in WBC, neutrophils, platelet, NLR, and SII, respectively. We also found that each 1% increase in 2-OHNAP was associated with a 1.11% (95% CI: 0.10%, 2.12%), 1.31% (95% CI: 0.018%, 2.63%), 1.11% (95% CI: 0.20%, 2.02%), and 1.82% (95% CI: 0.03%, 4.08%) increase in WBC, neutrophils, platelet, and SII, respectively. Each 1% increase in 9-OHFLU was associated with a 2.02% (95% CI: 0.50%, 3.56%), 2.63% (95% CI: 0.70%, 4.60%), and 2.33% (95% CI: 0.70%, 4.08%) increase in WBC, neutrophils, and monocytes, respectively (Figure 2).

### 3.5. Identification of the Key mPAE or OH-PAH Metabolites Based on WBC

The adaptive LASSO penalized regression model was used to select major driving chemical WBC with an aim to overcome the multicollinearity between chemicals. Finally, MOP, MEOHP, 1-OHNAP, and 4-OHPH were selected at the smallest λ value = 0.01885. At this point, the β coefficient was 0.017 for MOP, 0.0007 for MEOHP, 0.0008 for 1-OHNAP, and 0.007 for 4-OHPH (Figure 3). Subsequently, we employed these key four pollutants to explore their joint associations with blood cell-based inflammatory indicators.

### 3.6. Relationship between Mixture of OH-PAHs and mPAEs and Blood Cell-Based Inflammatory Indicators Based on BKMR

We further explored the overall associations of OH-PAHs and mPAEs with blood cell-based inflammatory indicators using BKMR models. Despite finding no significant associations between the mixture of OH-PAHs and mPAEs with WBC, lymphocytes and neutrophils (Figure 4A–C), a positive correlation trend still existed. Notably, there was a significant positive relationship between the OH-PAHs and mPAEs mixture and monocyte above the 50th percentile (Figure 4D). We further explored the single associations of the screened four key pollutants with the inflammatory indicators. We observed that MOP was significantly correlated with WBC, lymphocytes, neutrophils, and monocytes when other pollutants were set at the 25th, 50th, and 75th percentiles (Figure 5A–D). In addition, when the other OH-PAHs and mPAEs concentrations were kept constant at the median level, the single-dose response of MOP with WBC, neutrophils, and monocytes showed an increasing positive trend (Figure 6A–D). However, the chemical mixture showed no significant associations with platelet, NLR, PLR, and SII (Appendix A).

### 3.7. Quantile g-Computation to Explore the Relationship of Co-Exposure OH-PAHs and PAEs with Blood Cell-Based Inflammatory Biomarkers

As shown in Figure 7, Q-g models showed that the OH-PAHs and mPAEs mixture was associated with raised WBC (β: 0.061%, 95% CI: 0.025%, 0.097%), lymphocytes (β: 0.053%, 95% CI: 0.015%, 0.090%), neutrophils (β: 0.073, 95% CI: 0.026%, 0.120%), and monocytes (β: 0.069%, 95% CI: 0.027%, 0.110). Figure 7A showed that MOP, 1-OHNAP, and MEOHP (all weights > 10%) were the relatively predominant metabolites increasing WBC. Figure 7B showed that 4-OHPH, MOP, and MEOHP (weights > 10%) were relatively predominant metabolites associated with lymphocytes, while 1-OHNAP (weight > 10%) mainly contributed to the negative association with lymphocytes. Figure 7C showed that MOP, 1-OHNAP, and MEOHP (weights > 10%) were relatively major drivers for neutrophils. Figure 7D showed that 4-OHPH, MOP, and 1-OHNAP (weights > 10%) were relatively major metabolites driving monocytes, while MEOHP (weights > 10%) mainly contributed to the negative correlation with monocytes. However, the overall concentration of OH-PAHs and mPAEs showed no associations with platelet, NLR, PLR, and SII (Appendix A).

### 3.8. Sensitivity Analyses

To further validate the relationships between OH-PAHs or mPAEs and blood cell-based inflammatory indicators, we excluded 21 pregnant women with abnormal WBC levels (<4.0 × 10^9^/L and >12.0 × 10^9^/L). The remaining 297 early pregnant women were subjected to sensitivity analysis, and the metabolites of OH-PAHs or mPAEs remained closely associated with certain inflammatory indicators, implying the robustness of our results (Appendix A).

## 4. Discussion

Mothers are inevitably exposed to OH-PAHs and mPAEs due to their wide use [29] Co-exposure to OH-PAHs and mPAEs with inflammatory reactions, its key physiological process, and the achievement of homeostasis is essential for fetal growth and development. In the present study, we evaluated the links between the relationships of combined exposure of OH-PAHs and mPAEs with blood cell-based inflammatory indicators. General linear regression suggested the positive associations of MOP, 9-OHFLU, 4-OHPH, and 1-OHPH with WBC, neutrophils, lymphocytes, and monocytes. The BKMR model showed significant overall associations between the mixture of OH-PAHs and mPAEs with inflammatory markers; MOP is the leading chemical. Our study will provide an important basis for controlling pathological inflammatory states originating from environmental pollution in early pregnancy.

The urinary OH-PAHs and mPAEs were highly detected in enrolled pregnant women in southwest China, and the lowest detection rate is higher than 80%, which indicates high exposure to OH-PAHs and mPAEs for pregnant women in Guizhou. Notably, the median OH-PAHs and mPAEs concentration in our study was higher than that from other regions of China. For example, the median of MBP (96.13 μg/L) in the present study was far higher than that from Maanshan City (53.0 μg/L) [30]. In addition, compared to the median values of 1-OHNAP and 2-OHFLU among pregnant women (0.25 μg/L and 0.49 μg/L, respectively) in Tongji [31], our study showed relatively higher exposure doses (0.65 μg/L and 0.53 μg/L, respectively). The distinct patterns of pregnant women’s exposure to PAEs in different regions may be due to the disparity in dietary habits, use of PAA-containing products, and anthropogeographic nature [21,32]. The widespread use of plastic films, fertilizers, and pesticides in agriculture may contribute to the high PAEs levels in the tobacco-soil system in Guizhou [33,34], which contributes to a large quantity of detection of DBP, DMP and DEHP in soil that is metabolized in the body to mPAEs [35]; this may explain the high exposure to mPAEs in our study [33]. Moreover, the thinness and high permeability of soils in the Karst topography of Guizhou lead to the vertical transfer of PAHs from soils into groundwater, causing more serious water-born PAHs pollution [36]. Therefore, the geographical environment and the economy and culture differs greatly from that in eastern China, which may partially explain the widespread exposure to PAHs and PAEs in Guizhou. However, there are few studies addressing its adverse health association based on the local specific exposure pattern. Our study can provide scientific information for maternal health services in Guizhou.

Numerous studies have explored the relationships between PAHs or PAEs and inflammatory status in men or children [12,16]. Furthermore, pregnant women appear more susceptible to environmental pollutants, as well as pregnancies is accompanied by profound changes in the cardiovascular, respiratory, and endocrine functions [37]. Kelly K. Ferguson et al. found that urinary PAHs metabolites of late pregnant women were positively associated with inflammatory cytokines and risk of oxidative stress, but there are few studies on early pregnancy [38]. Zhan et al. also explored the association of PAHs with hematologic changes among late pregnant women, but they only determined several cell-based inflammatory markers (WBC and Platelet) [39]. Under our present study, we systematically explored the relationships between PAHs and PAEs mixture exposure with blood cell-based inflammatory indicators (WBC, lymphocyte, neutrophiles, monocyte, platelet, NLR, PLR, and SII). Our study revealed that OH-PAHs and mPAEs co-exposure was linked with elevated WBC, neutrophils, lymphocytes, and monocytes in the first trimester. Considering that early pregnancy is a vulnerable period of exposure to phthalate metabolites (MEP, MnBP, MECPP, MEHHP, and MEOHP) [40] and the regulation of inflammatory expression in early life phase is essential for fetal survival [41], therefore, addressing the inflammatory level associated with environmental pollutants on first-trimester stage has great significance for embryo development and maternal health.

Moreover, the co-exposure of OH-PAHs and mPAEs has been a concern due to humans being commonly exposed to a mixture of OH-PAHs and mPAEs in real environmental scenarios. However, most current studies separately explored the adverse effects of OH-PAHs or mPAEs exposure on inflammatory indicators [17,42,43]. In contrast, we explored co-exposure to OH-PAHs and mPAEs, considering the realistic exposure scenarios and the results showed that co-exposure to OH-PAHs and mPAEs was also associated with elevated inflammatory indicators, and MOP acts as a major contributor. We also used the adaptive LASSO penalty regression model to counteract the multicollinearity of multiple OH-PAHs and mPAEs metabolites due to the strong correlation between OH-PAHs and mPAEs metabolites; and the most important OH-PAHs or PAEs metabolites associated with inflammatory indicators were selected. We first employed the selected key pollutants to explor its combined association with blood cell-based inflammatory indicators based on the LASSO model.

Although the biological mechanisms triggering co-exposure of PAHs and PAEs by increasing the hemocyte-based inflammatory markers remain unclear, numerous studies have provided the clue that serum cytokines may function as an inter-mediator [12,17,44]. Pre-treated macrophages by PAHs in vitro show pro-inflammatory cytokines (e.g., IL-6, IL-12) [45]. Mechanistically, PAH binds and actives AhR, which primes the IL-6 promoter by binding to the its dioxin response element, and these processes facilitate IL-1-induced activated tumor necrosis factor-β (TNF-β) components, which ultimately contribute to IL-6 production [46]. Moreover, PAEs tend to stimulate IL-6 and IL-8 production, and IL-6 stimulation is more potent [47]. Hence, the raised IL-6 may account for balanced inflammatory indicators originated from PAHs or PAEs exposure. According to certain research, the most well-established mechanism of PAHs’ downstream toxicity is oxidative DNA damage, which causes adverse inflammatory symptoms as a result of oxidative stress. When PAH receptors are bound, oxidative stress results, and higher levels of 8-OHdG and 8-iso are linked to exposure to PAHs as determined by urine measurement of hydroxylated metabolites, suggesting that females may be more vulnerable to this harm [39,48]. However, the effect of the PAH and PAE co-exposure on the inflammatory cytokines in vivo and in vitro is lacking.

We first explored the overall association of PAHs and PAEs with blood cell-based inflammatory indicators in early pregnant women. However, our study has several limitations. Firstly, the causal association between co-exposure to PAHs and PAEs and hemocyte-based inflammatory indicators can not be clearly clarified by this cross-sectional methodology. Secondly, no assessment was made of the interconnections between other typical environmental contaminants, such as metals and particulate matter. Thirdly, we failed to repeatedly examined the PAHs and PAEs, because the dot urine samples only reflect the instantaneous exposure of PAEs and PAHs. Therefore, further studies are needed to continuously detect PAHs and PAEs in urine and explore its mixed associations with blood cell-based inflammatory indicators. Forthly, residual confounds, such as diet patterns should be taken into consideration in further studies. However, there are some advantages of the present study. The combined association of PAHs and PAEs on health outcomes is of increasing interest, as humans are simultaneously exposed to multiple PAHs and PAEs. Determining the association of PAHs and PAEs mixture with blood cell-based inflammatory indicators could realistically reflect their hazardous association. Secondly, we focused on the association of PAHs and PAEs mixtures with blood cell-based inflammatory indicators at the early stage of pregnancy, which is a particularly sensitive and vulnerable window to environmental exposure. Thirdly, we first determined that the above association among pregnant women in southwest China is apparently different from another territories due to the anthropogeography. We added the scientific evidence that co-exposure to PAHs and PAEs was associated with elevated hemocytes-based inflammatory indicators.

## 5. Conclusions

We found that increased inflammatory indicators during the first trimester were linked to co-exposure to OH-PAHs and mPAEs, among which MOP might be the major contributor. Therefore, our study provides scientific information for perinatal health service to reduce and control the maternal inflammatory state-associated adverse perinatal outcomes caused by environmental pollution, especially for early pregnant women. Future studies regarding the association between OH-PAHs and mPAEs mixtures with blood cell-based inflammatory indicators at second and third trimester stage should be undertaken. Moreover, the exposure window effect also needs to be determined.

## Figures and Tables

**Figure 1 toxics-11-00810-f001:**
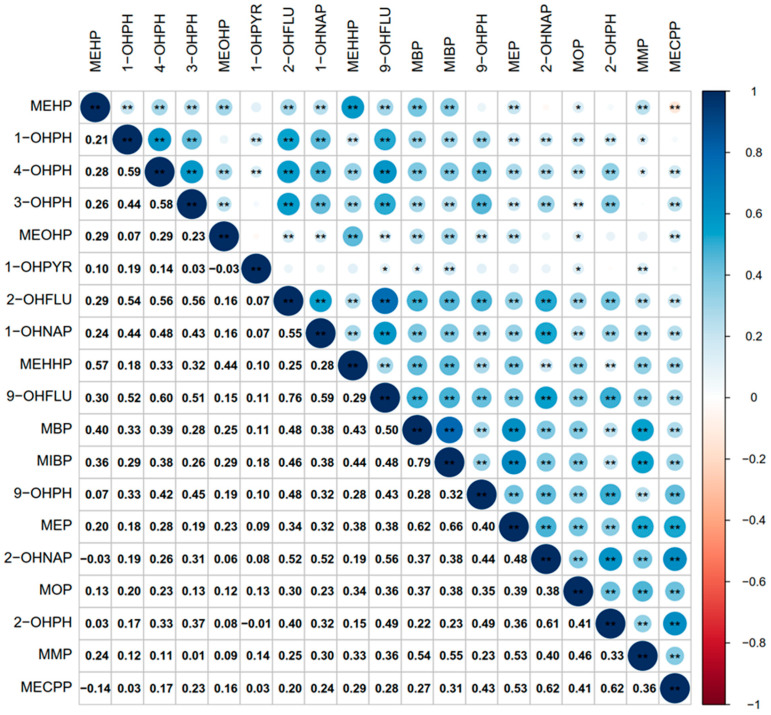
Spearman correlation coefficient between OH-PAHs and mPAEs in urine. * *p* < 0.05, ** *p* < 0.01.

**Figure 2 toxics-11-00810-f002:**
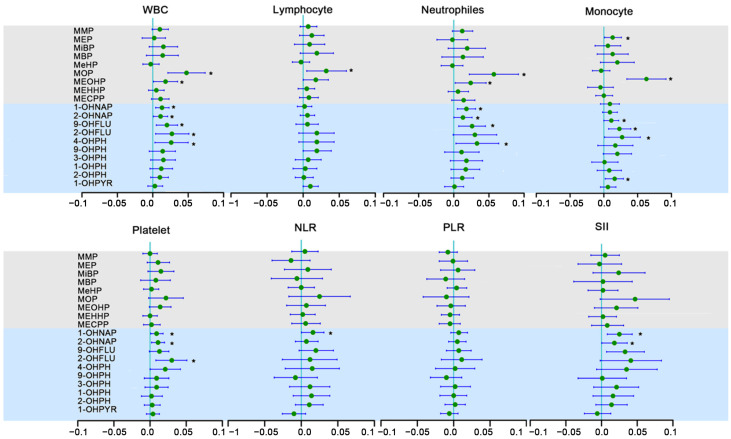
Association of OH-PAHs and PAEs with blood cell-based biomarkers of inflammation in the urine of pregnant women. Note: Linear regression models are adjusted for maternal age, pre-pregnancy BMI, marital status, categorical education, race, smoking, passive smoking, exercise, and alcohol consumption. * *p* < 0.05.

**Figure 3 toxics-11-00810-f003:**
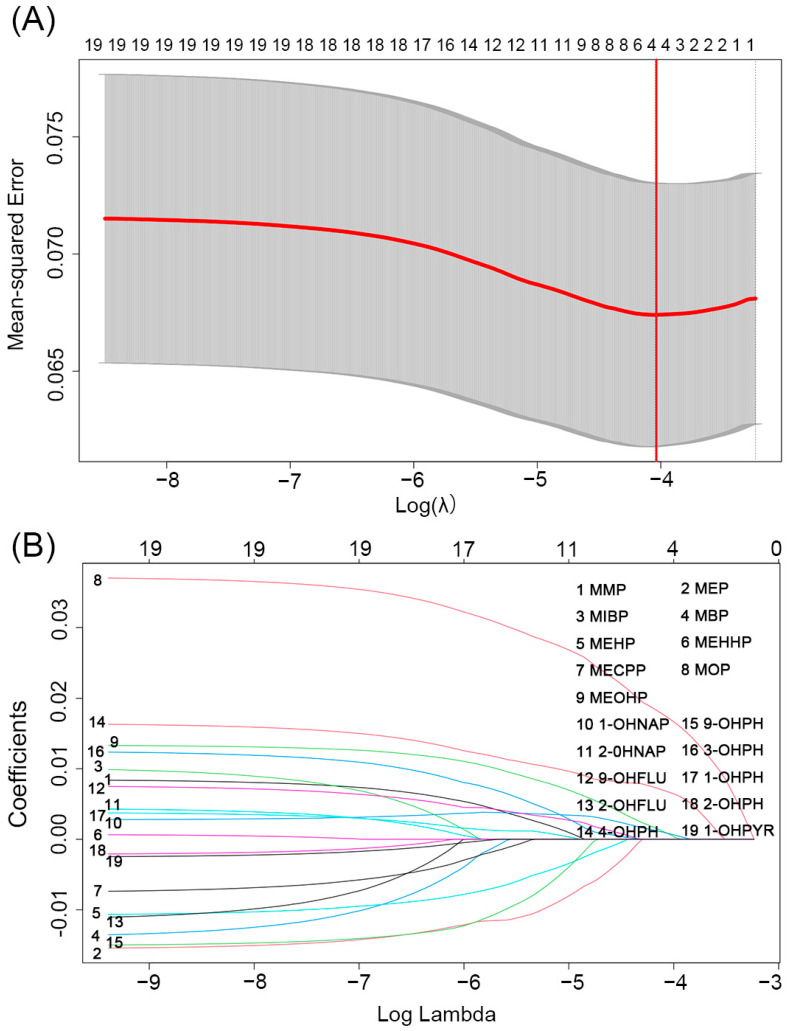
Adaptive lasso penalty regression was used to analyze the relationship between phthalate and PAH metabolite exposure and WBC. Note: WBC was the dependent variable and OH-PAHs and PAEs metabolites were the independent variables in the lasso penalty regression model. The model adjusted for maternal age, pre-pregnancy BMI, marital status, categorical education, race, smoking, passive smoking, exercise, and alcohol consumption. The red line in (**A**) indicates λmin. Coefficient curves for OH-PAHs and m-PAEs are shown in (**B**). (for the interpretation of the curves in this figure, the reader is referred to the labeling in the figure).

**Figure 4 toxics-11-00810-f004:**
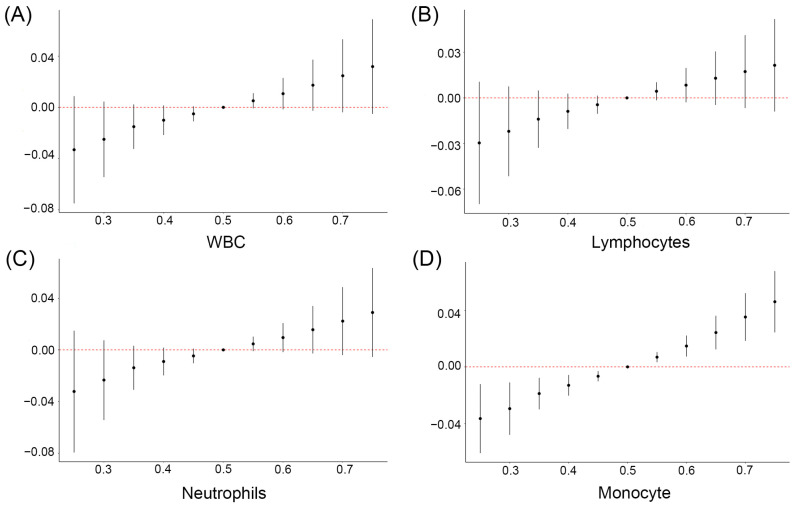
The overall association of screened OH-PAHs and PAEs mixtures with WBCs (**A**), lymphocytes (**B**), neutrophils (**C**), and monocytes (**D**) was estimated by the Bayesian kernel machine regression (BKMR) model. Note: The model adjusted for maternal age, pre-pregnancy BMI, marital status, categorical education, race, smoking, passive smoking, exercise, and alcohol consumption.

**Figure 5 toxics-11-00810-f005:**
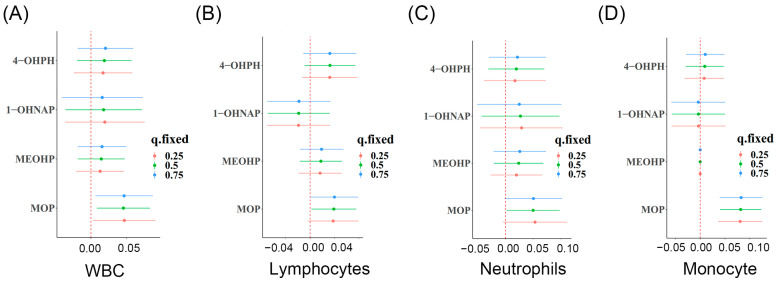
The single associations of screened OH-PAHs and PAEs with WBCs (**A**), lymphocytes (**B**), neutrophils (**C**), and lymphocytes (**D**) were estimated by the Bayesian kernel machine regression (BKMR) model. Note: The model adjusted for maternal age, pre-pregnancy BMI, marital status, categorical education, race, smoking, passive smoking, exercise, and alcohol consumption.

**Figure 6 toxics-11-00810-f006:**
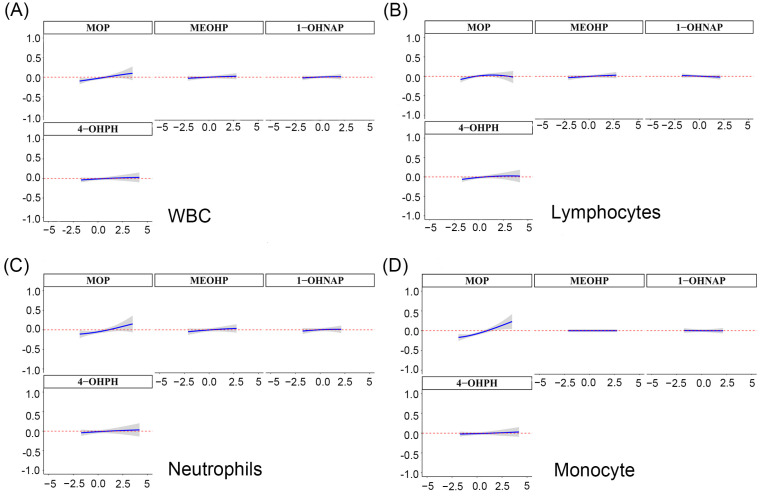
Univariate exposure-response relationships between the concentration of each substance and WBCs (**A**), lymphocytes (**B**), neutrophils (**C**), and monocytes (**D**) when the other substances were fixed at median concentrations. Note: The model adjusted for maternal age, pre-pregnancy BMI, marital status, categorical education, race, smoking, passive smoking, exercise, and alcohol consumption.

**Figure 7 toxics-11-00810-f007:**
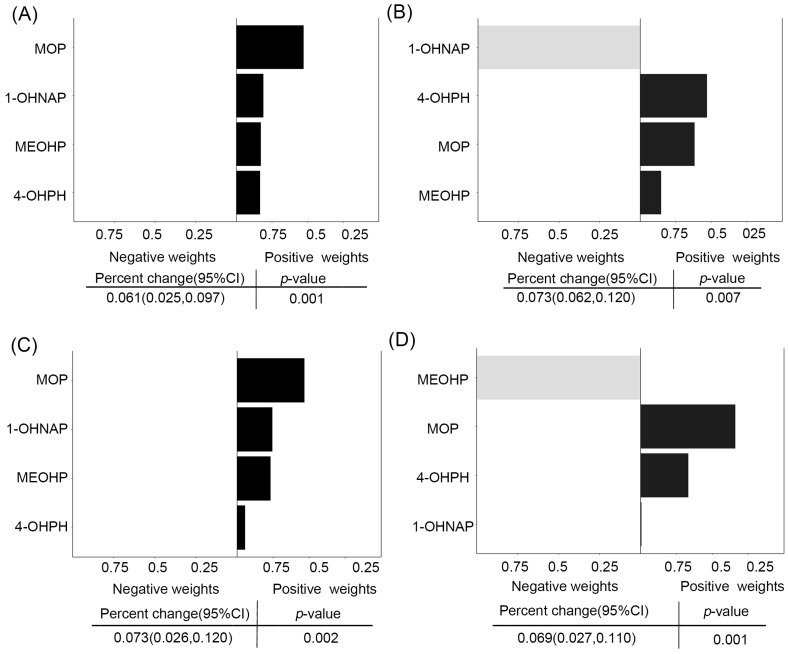
Estimation of the effect of screening the four mixtures of OH-PAHs and PAEs in WBCs (**A**), lymphocytes (**B**), neutrophils (**C**), and monocytes (**D**) and scaled weights corresponding to the proportion of the effect for each chemical in quantile g-computation. Note: The model adjusted for maternal age, pre-pregnancy BMI, marital status, categorical education, race, smoking, passive smoking, exercise, and alcohol consumption.

**Table 1 toxics-11-00810-t001:** Baseline characteristics of the pregnant women (*n* = 318).

Characteristics	*n*	Mean ± SD or Percent (%) or Median (Lower Quartile, Upper Quartile)
Age	318	26.22 ± 4.7
Pre-pregnancy BMI (kg/m^2^)	318	22.10 ± 4.22
Ethnicity		
Ethnic Han	312	98.11
Minority	6	1.8
Education level		
High school or below	232	72.95
College or beyond	86	27.04
Marital status		
Married	277	87.11
Single	39	12.26
Divorced	2	0.62
Smoking	20	6.28
Passive smoking	313	98.43
Activities		
Little or no	167	52.52
1–2/week	41	12.89
≥3/week	110	34.59
Drink	27	8.49
WBC	318	8.09 (6.80, 9.58)
Lymphocyte	318	1.80 (1.50, 2.18)
Neutrophile granulocyte	318	5.73 (4.60, 6.94)
Monocyte	318	0.40 (0.32, 0.48)
Platelet	318	241.5 (202, 274)
NLR	318	3.11 (2.43, 3.94)
PLR	318	130.50 (109.44, 155.56)
SII	318	727.00 (552.05, 942.74)

Abbreviations: BMI, body mass index; SD, standard deviation; WBC, white blood cell count; NLR, neutrophil to lymphocyte ratio; PLR, platelet to lymphocyte ratio; SII, systemic immune-inflammation index.

**Table 2 toxics-11-00810-t002:** Distributions of urinary OH-PAHs and mPAEs (ng/mL).

Urinary Metabolites	>LOD (%)	Median (25th, 75th)	Creatinine Correction Median (25th, 75th)
1-OH-NAP (μg/L)	70.05	0.65 (<LOD, 2.06)	0.47 (0.006, 1.56)
2-OH-NAP (μg/L)	82.03	1.00 (0.10, 4.42)	1.09 (0.10, 3.20)
9-OH-FLU (μg/L)	91.15	0.39 (0.16, 0.81)	0.29 (0.13, 0.60)
2-OH-FLU (μg/L)	99.22	0.53 (0.25, 1.04)	0.44 (0.19, 0.85)
4-OH-PHE (μg/L)	62.5	0.03 (<LOD, 0.07)	0.023 (0.009, 0.05)
9-OH-PHE (μg/L)	86.46	0.02 (0.008, 0.07)	0.019 (0.007, 0.06)
3-OH-PHE (μg/L)	95.57	0.30 (0.10, 0.77)	0.22 (0.08, 0.71)
1-OH-PHE (μg/L)	83.59	0.08 (0.01, 0.24)	0.06 (0.01, 0.17)
2-OH-PHE (μg/L)	77.34	0.09 (0.02,0.55)	0.07 (0.01, 0.34)
1-OH-PYR (μg/L)	74.22	0.01 (<LOD, 0.06)	0.01 (0.0002, 0.04)
MMP (μg/L)	68.75	2.15 (<LOD, 6.99)	1.48 (0.03, 4.91)
MEP (μg/L)	97.66	17.55 (6.92, 37.30)	13.17 (5.42, 31.91)
MIBP (μg/L)	99.22	19.59 (9.74, 36.56)	14.52 (7.28, 27.01)
MBP (μg/L)	100	96.13 (39.29, 175.50)	64.23 (28.92, 144.39)
MEHP (μg/L)	78.39	3.69 (0.74, 8.87)	2.30 (0.29, 7.96)
MOP (μg/L)	65.63	0.12 (<LOD, 0.25)	0.85 (0.04, 0.17)
MBZP (μg/L)	40.25	0.03 (<LOD, 0.15)	0.03 (0.01, 0.11)
MEOHP (μg/L)	88.54	10.84 (4.07, 38.97)	9.67 (3.67, 32.53)
MEHHP (μg/L)	90.36	12.77 (5.07, 25.91)	8.75 (3.10, 21.69)
MECPP (μg/L)	78.12	130.40 (32.19, 2490.23)	134.74 (23.58, 2372.43)

Mono-methyl phthalate (MMP), mono-ethyl phthalate (MEP), mono-isobutyl phthalate (MIBP), monobutyl phthalate (MBP), mono-octyl phthalate (MOP), monobenzyl phthalate (MBZP), mono(2-ethylhexyl) phthalate (MEHP), mono(2-ethyl-5-oxohexyl) phthalate (MEOHP), mono (5-carboxy-2-ethylpentyl) phthalate (MECPP), mono(2-ethyl-5-hydroxyhexyl) phthalate (MEHHP). PAH metabolites were 1-naphthol (1-OH-NAP), 2-naphthol (2-OH-NAP), 2-hydroxyfluorene (2-OH-FLU), 9-hydroxyfluorene (9-OH-FLU), 1-hydroxy-phenanthrene (1-OH-PHE), 2-hydroxyphenanthrene (2-OH-PHE), 3-hydroxyphenanthrene (3-OH-PHE), 4-hydroxyphenanthrene (4-OH-PHE), 9-hydroxyphenanthrene (9-OH-PHE), and 1-hydroxypyrene (1-OH-PYR).

## Data Availability

The 318 pregnant women analyzed were all from the birth cohort database of Guizhou Province, and reasonable requests could be obtained from the corresponding authors.

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
