# Peer review of "Co-Exposure of Polycyclic Aromatic Hydrocarbons and Phthalates with Blood Cell-Based Inflammation in Early Pregnant Women"

_toxics, 2023, doi:10.3390/toxics11100810_

Round 1
Reviewer 1 Report
General comments: Polycyclic aromatic hydrocarbons (PAHs) and phthalates (PAEs) are the ubiqutious environmental chemicals that can lead to multiple adverse health outcomes. Inflammation induced by these chemicals may be a proposed mechanism. This study examined the assoications between individual and mixture of PAHs and PAEs and blood cell parameters. The topic is improtant and have multiple statistical approaches applied. However, there are some issues that should be addressed. My specific comments are listed below.
1) The author used "effect" throughout the manuscript. However, given the observational nature of the study, only association can be inferred rather than a causal relationship. Please revise it throughout the paper.
2) The abbreviation should be presented by its full-name when it occurs firstly. Please check this issue throughout the paper.
3) Line 33: The words “linked to” should be correted as “linked with” given the observational nature of the study.
4) Lines 85-87: Are there any references to back up the statement?
5) Please provide more details on the study population. e.g., How many subjects were included and excluded?
6) The timing and interval of urine and blood collection, as well as the conditions of fasting prior to collection, were not clearly explained.
7) Please add more details about the quality control of chemical determinations including the spiked recoveries, variation and limits of detection.
8) The authors should clarify how to select potential confounders in regression models.
9) Line 365: Does the pesticides used contribute to exposure to mPAEs?
10) In addition to altered serum cytokines as a potential biological mechanisms, are there any others? Such as oxidative stress?
11) Line 357-371: The authors described four OH-PAHs and mPAEs co-metabolites associated with elevated leukocytes, lymphocytes, neutrophils, and monocytes based on the Q-g model. However, platelet, NLR, PLR, and SII were not mentioned as markers and are suggested to be added.
The English language should be improved.
Reviewer 2 Report
I have some major queries of this work that would need to be clarified prior to publications.
1. The authors discuss how certain women were excluded from the study. For those with serious infectious disease etc (L95-97), this seems entirely reasonable, however, I believe the number that were excluded based on this should be included. The next exclusion, based on "inflammatory disease at baseline" (L97-99) reads like their baseline blood counts were taken, they were higher than others, and so were excluded. The authors need to clarify if these women had a pre-diagnosed inflammatory condition, or if the baseline was used as a means to diagnose the condition? Again, the number excluded should be included. Finally, from my reading of the methods, 318 women were initially included in the study, and then certain individuals were removed from the study due to the above mentioned conditions. But in Table 1, the n provided for the blood counts is 318, which would suggest no women were removed from the study? Do the means used in Table 1 come from an n of 318, or is it different? L117 in the methods would suggest this is the n, but then where do the exclusions come from? Further to this, L175 mentions that some women were excluded for aberrant WBC. Does this refer to the "inflammatory condition" mentioned previously, or is this separate? Again, no numbers are provided on how many this was, and whether these were originally included in the n of 318.
2. The authors use the term "blood cell-based inflammatory biomarkers" or a derivative of it, many times in the manuscript. While NLR, PLR and SII may be considered inflammatory markers in some clinical settings, levels of individual cell types (lymphocyte etc) are far less used. This is important as in Figure 2, there is far less significant association between OH-PAEs/PAEs levels and NLR/PLR/SII than there is to the individual cell counts. MOP is highlighted by the authors as a "major contributor(s) to this association", however is not significantly associated with NLR/PLR/SII levels, only the WBC and individual cell types. Further to this, it is unusual to use the term "biomarker" for these measurements, which would more likely be C-reactive protein (CRP), erythrocyte sedimentation rate (ESR) and plasma viscosity (PV). I think the authors should remove usage of "biomarker" from the text, and also clearly define what they mean by "blood cell-based inflammation", which is the term used in the title.
3. There are lots of sentences that need to be re-worked so that the language makes sense. It makes it quite difficult to assess the validity of many of the claims, when the language is not correct. There are too many examples to list, I think the whole manuscript needs to be worked through to ensure that the language is correct and accessible for any readers.
4. Why is educational level or marital status included in this analysis? Particularly marital status seems entirely irrelevant to this study and the authors do not provide any further mention of these groups, beyond their initial mention in Table 1.
5. The mass spec part of the method is just cited as another paper(s), with no details given here on the methodology. This is the major methodology used in this paper, and yet not fully detailed here. This needs to be rectified.
Minor points:
1. The first sentence of the introduction needs re-working as it reads like PAHs and PAEs are naturally occurring.
2. Abbreviations are either not defined, or frequently defined after their initial use. I cannot find that LOD is ever defined,
3. Fig 1-3 need to be made larger, I find them difficult to read.
4. Each legend seems to re-define the abbreviations. This isn't necessary.
Mentioned above, as this is a major point that needs to be re-worked.
Round 2
Reviewer 1 Report
The authors have addressed my comments and I would like to suggest pulishing the manuscript.
NO.
Reviewer 2 Report
Thank you for engaging with feedback, I would be happy for this to be published.
Issues addressed.